# Local Governments Spending on Promoting Physical Activity during 2015–2020: Financial Data and the Opinion of Residents in Poland

**DOI:** 10.3390/ijerph191912798

**Published:** 2022-10-06

**Authors:** Karolina Sobczyk, Mateusz Grajek, Mateusz Rozmiarek, Krzysztof Sas-Nowosielski

**Affiliations:** 1Department of Health Economics and Health Management, School of Health Sciences in Bytom, Medical University of Silesia in Katowice, 41902 Bytom, Poland; 2Department of Public Health, School of Health Sciences in Bytom, Medical University of Silesia in Katowice, 41902 Bytom, Poland; 3Department of Sports Tourism, Faculty of Physical Culture Sciences, Poznan University of Physical Education, 61871 Poznan, Poland; 4Department of Humanistic Foundations of Physical Culture, Faculty of Physical Education, Jerzy Kukuczka Academy of Physical Education in Katowice, 40065 Katowice, Poland

**Keywords:** physical activity, recreation, sports services, local government, COVID-19

## Abstract

Introduction: The COVID-19 pandemic has significantly affected local governments involved in sports and recreation in designated areas. The unprecedented scale of the spread of the disease has led to increased research in the area of the disease, considering various correlations. However, little has been written about the impact of the pandemic on local government spending on recreation and sports services in Poland. Objective: The purpose of the article is to assess the level of local government involvement in the implementation of sports and recreation in Poland compared to other European Union countries, as well as changes in this level in connection with the emergence of the COVID-19 pandemic in the opinion of respondents. Methodology: In the study, the data regarding expenditures of local government units on recreational and sporting services collected in the Statistical Office of the European Union (EURO-STAT) for 2015–2020 were used. The survey portion was conducted among 1600 respondents who provided answers on a 5-item scale that addressed local government involvement in promoting physical activity among residents during COVID-19. Results and conclusion: Local government spending on recreational and sporting services in Poland between 2015 and 2019 increased by about 38%, from EUR 1524.7 million in 2015 to EUR 2103.5 million in 2019. This spending in 2019 was about 40% higher than the average for European Union countries. In contrast, in 2020 it amounted to more than EUR 1886 million and was more than 10% lower compared to the previous year (2019)—the pre-pandemic period. Despite the obstacles caused by the COVID-19 pandemic and budgetary constraints, cities in Poland took several measures in 2020 to maintain the current pace of development and strived to maintain the status of modern, green, and open, betting on balanced development also in aspects related to sports or culture. It was shown that the opinion of respondents mostly coincided with the existing financial state—in voivodeships where there had been a decrease in spending related to sports and recreation compared to the pre-pandemic period, residents are worse at assessing the activities of local governments related to promoting physical activity.

## 1. Introduction

The World Health Organization (WHO) announced a global pandemic caused by the severe acute respiratory syndrome coronavirus (SARS-CoV-2), which turned into a public health emergency of global significance, on 11 March 2020 [1]. To combat the transmission of coronavirus, social isolation and confinement are essential. The country-wide lockdown, on the other hand, had never happened before, and it was unclear how it would affect people’s health and well-being. In these situations, the sudden and stressful scenario, along with extended stays at home, may result in a radical shift in lifestyle behavior, including physical activity (PA), eating habits, alcohol use, mental health, sleep quality, etc. [2,3,4,5,6,7].

Similarly, there was widespread worry about the harmful effects of inactivity and sedentary behavior on one’s health [8]. The general recommendation for considering an adult to be physically active is to spend at least 150 min of moderate or 75 min of vigorous-intensity activity per week or an equivalent combination of both [9], and sedentary behavior is defined as any waking behavior practiced while lying down, reclining, sitting or standing, involving an energy expenditure of ≤ 1.5 metabolic equivalents [9]. While the disease spread worldwide, healthy people were asked to stay at home for extended periods, and as a result, COVID-19 dramatically changed the determinants of both types of behavior (individual, interpersonal, environmental, regional or national policy, and global) [10]. Due to isolation and limitations on engaging in communal activities, adhering to PA recommendations and minimizing sedentary behaviors during the lockdown were difficult, especially in the first few weeks when the population had few opportunities to remain active even at home [11].

The rapid spread of the COVID-19 pandemic in Europe and the increasingly stringent measures taken by authorities to prevent the spread of the disease had profound effects on the healthcare system and the overall European economy [12]. The strategies of European governments during the COVID-19 pandemic were aimed at ensuring the long-term sustainability of health systems in the face of limited intensive care unit capacity and equipment, as well as supporting the economy by providing cash, loans, and guarantees to compensate for losses incurred because of pandemic restrictions [13,14].

Establishing partnerships with traditional and non-traditional physical activity providers is an important part of promoting health-related physical activity at the community level. Local government, in particular, contributes significantly to the health and well-being of the community by contributing to infrastructure and the built environment [15]. Local health departments are well-positioned to act as catalysts for the institutional and community reforms that are required to boost physical activity among the general public. Evidence-based strategies, such as the promotion of high-quality physical education in schools, social support networks and structured programs for physical activity in communities, and workplace practices, policies, and programs that promote physical activity should be prioritized [16,17]. In communities where the built environment poses considerable hurdles to physical exercise, particularly in economically disadvantaged populations disproportionately burdened by chronic disease, health departments must also focus on land use and transportation practices and laws [18,19].

The purpose of this paper is to assess the level of involvement of local governments in the implementation of sports and recreation tasks in Poland compared with other European Union countries, as well as changes in this level in connection with the COVID-19 pandemic. In addition, the purpose of the publication is to assess the level of involvement of local governments in Poland in the implementation of sports tasks in individual voivodeships. Voivodeships are areas administered by a voivode (governor) in Poland and several other central and eastern Europe countries.

We assessed the level of involvement based on local government budget expenditures incurred for this purpose and the opinion of residents measured on a 5-point scale. The paper answers two key research questions:How do the expenditures of Polish local governments on sports and recreation compare to other voivodeships of Poland and to the European Union?How do residents evaluate the above activities during the COVID-19 pandemic?

## 2. Materials and Methods

To realize the research aim, local budget government expenditures on recreational and sporting services were analyzed, taking into consideration the following: (1) In European Union countries: the level of expenditures, the share of expenditures in total expenditures, expenditures by percentage of gross domestic product (GDP); (2) In Poland: the level of expenditures, mean value of expenditures per inhabitant.

In the study, the data regarding local government expenditures on recreational and sporting services from the Statistical Office of the European Union (EUROSTAT) for 2015–2020 were used. According to the classification of functions of government (COFOG), expenditures on recreational and sporting services were classified in Division GF0801—recreational and sporting services [16]. Moreover, the study analyzed data from the Central Statistical Office’s Local Data Bank on local government spending on physical culture from 2017 to 2020 in Poland. Data for 2021 were unavailable at the time of the survey, as these are published with a two-year delay.

Physical culture expenses were classified under Division 926—Physical Culture, according to the Polish budget classification system [17]. The following chapters are listed in this division: 92,601—sports facilities, 92,604—physical culture institutions, 92,605—tasks in the scope of physical culture, 92,678—elimination of the effects of natural disasters, 92,679—foreign support, 92,680—research and development activity, and 92,695—other activity. For analysis, the years presented were grouped: the period before 2020 was called the pre-pandemic period, and 2020 was called the COVID-19 pandemic period.

In addition, the survey included an anonymous questionnaire survey using the indirect survey method through online forms (CAWI). The survey involved 1600 people (100 from each voivodeship of Poland) who met the inclusion criteria, i.e., they were Polish citizens, were 18 years of age or older, and declared that they were familiar with local government projects for the promotion of physical activity among residents.

The research tool was a 5-item author’s scale, in which participants marked their agreement with a given statement on a 5-point Likert scale: strongly agree, rather agree, have no opinion, rather disagree, strongly disagree. The statements on the scale were scored as follows: full agreement—5 points, no full agreement—1 point. The point totals were counted, and arithmetic means standard deviations, and other descriptive statistics were calculated on their basis. The internal reliability of the scale was estimated at α = 0.82. In addition, the scale was previously validated on a test sample of 30 people. Validation testing was carried out twice (at an interval of two weeks). Based on the validation, the scale was found to be consistent, and its results were reproducible (*p* = 0.001).

The data were also subjected to statistical inference. A Kruskal–Wallis test with an epsilon-square indicator for the strength of the relationship were used. To show differences between spending levels in different regions, we examined the differences between the scores obtained on the scale and the state of the budget during the COVID-19 pandemic. For this purpose, the chi-square test and the V-Cramer coefficient of relationship strength (with Yates and Fisher’s correction) were used. The probability level was set at *p* = 0.05.

## 3. Results

Local government spending on recreational and sporting services in Poland between 2015 and 2019 increased by about 38%, from EUR 1524.7 million in 2015 to EUR 2103.5 million in 2019. This spending in 2019 was about 40% higher than the average for European Union countries. In contrast, 2020 expenditures amounted to over EUR 1886 million and were more than 10% lower compared to the previous year (2019)—the pre-pandemic period. Decreases in local government spending on recreation and sport in the 2019–2020 period were recorded in 16 of the 24 European Union countries for which the analyzed data is available. The largest decreases in spending were reported in Lithuania, Hungary, Belgium, and Poland. Taking into account the amount of local government spending on recreation and sport in 2020, Poland was in fifth place, after France, Spain, the Netherlands, and Sweden, with a value above the EU average. The lowest spending was recorded in Cyprus, Latvia, and Bulgaria—T = 13.421; *p* = 0.0003. Local government spending on recreational and sporting services in Poland accounts for only 2.4% of total spending at this level of government. This value is well below the average for the European Union (4%) and placed Poland in 17th place among the other countries for which data is available. This value in Poland remained at a similar level in 2015–2019, and in 2020 it decreased by 0.4% compared to the previous year. Decreases in this area were recorded overall in 18 of the 24 countries for which data are available. The highest percentage of total expenditure on recreational and sporting services was spent by local governments in Greece, Cyprus, and Luxembourg. The lowest was in Italy, Denmark, and Ireland—T = 12.331; *p* = 0.0002 (Table 1).

In Poland, between 2015 and 2020, excluding 2016, the percentage of GDP allocated to recreational and sporting services remained constant (0.4%). This is higher than the average for European Union countries for which this data is available (0.3%). The highest percentage of GDP was spent on recreational and sporting services in Sweden, followed by Greece, France, the Netherlands, and Finland. The lowest were Slovakia, Italy, and Bulgaria—T = 11.181; *p* = 0.0001.

In the analyzed period in Poland, one may observe systematic growth of local government spending on physical culture. In 2017–2020, these expenditures increased from EUR 27.29 million to EUR 42.53 million, i.e., by 55.9%. The period 2019–2020 also saw an increase of 5.27%. The largest increase was in Podlaskie and Lower Silesia voivodeships. Spending on physical culture in the year of the COVID-19 pandemic outbreak decreased compared to the pre-pandemic year (2019) in local governments of 10 out of 16 voivodeships. The highest expenditures on physical culture per capita were recorded in 2020 in Lower Silesia, Lubuskie, and Upper Silesia voivodeships; the lowest were in Podkarpackie, Swietokrzyskie, and Pomorskie voivodeships—T = 11.546; *p* = 0.0001 (Table 2).

Based on the scale used, we compared respondents’ opinions on local government involvement in promoting physical activity (sports and recreation) and the difference in spending before and during the COVID-19 pandemic. It was observed that wherever spending on sports and recreation increased between 2019–2020, the opinions were better; where spending decreased, they were worse. The relationship was confirmed statistically for most regions (*p* < 0.05) except Pomorskie and Zachodniopomorskie (*p* > 0.05). Residents of Podlaskie voivodeship rated physical activity activities best (4.32 points) and reported the highest increase in spending on sports and recreation (+86.68)—X^2^ = 9.786; V = 0.498; *p* = 0.0001. On the other hand, these activities were rated the worst in the Swietokrzyskie voivodeship (2.16 points), which recorded the second-lowest decrease in spending on physical activity (−23.91)—X^2^ = 10.764; V = 0.698; *p* = 0.0001. Detailed data and analysis are shown in Table 3.

## 4. Discussion

The COVID-19 pandemic significantly affected local governments involved in sports and recreation in designated areas [12]. The unprecedented scale of the spread of the disease led to increased research in the area of the disease. However, little has been written about the impact of the pandemic on local government spending on recreation and sports services in Poland. The finances of local government units are an extremely important element of public funds management, as their condition affects the entire economy of the country [2,3,4,5,6,7].

The threat of the SARS-CoV-2 virus affected the economic and social situation, as well as the activities of public administration at every level [20]. According to the analysis of the obtained results, the COVID-19 pandemic contributed to a decrease in local spending on recreation and sports in many European Union countries. The largest appeared in Central and Eastern Europe, including Poland, which before the pandemic had consistently raised the pool of funds for sports and recreational services since 2015. Before the pandemic, the country remained at the forefront of European countries, showing local government spending significantly above the European Union average. The percentage of GDP spent on sports and recreation was also above the EU average. In contrast, during the pandemic, local government spending on recreational and sporting services was below the EU average and placed Poland in 17^th^ place among other countries for which data was available. It should also be noted that strengthening popular and competitive sports and even developing sports infrastructure throughout Poland, as well as encouraging physical activity, remain among the main goals of the Ministry of Sports and Tourism, despite the COVID-19 pandemic. In 2021, a record budget of PLN 282 million was allocated for the development of Polish sports, as well as an additional PLN 117 million per year under the Physical Culture Development Fund for the physical activity of Poles [21].

Despite the obstacles caused by the COVID-19 pandemic and budgetary constraints, cities in Poland in 2020 took several measures to maintain their pace of development and strived to maintain their status as modern, green, and open, also betting on sustainable development in aspects related to sport or culture [21,22,23,24]. However, the present study results indicating decreased spending on sports and physical activity after the COVID-19 pandemic compared with the pre-pandemic year were confirmed [21,22].

The possibilities of sports and physical activity for Poles have been significantly limited because of COVID-19 restrictions. However, local governments proposed numerous alternatives to events or other sports and recreational activities canceled due to the pandemic. In Poznan, for example, the PKO Poznan Virtual Run was held individually, with participants signing up remotely. The data from the city report indicate that as many as 1619 people participated in the run [21]. Moreover, running, due to its limited contact with other people, turned out to be one of the safest physical activities during the COVID-19 pandemic, not only in Poznań itself, but also in the entire region surrounding the city, which was confirmed by the study performed by Rozmiarek and colleagues [25], who indicated the popularity of running at night in the natural environment to facilitate falling asleep and improve sleep quality. Similar searches for alternative, safe, and economically viable methods of practicing sport were also conducted in other cities, as indicated in the study by Mucha and Mucha [26] highlighting the popularity of extending the scope of training with Internet solutions, such as online training or remote consultations with sports experts, which was also confirmed in the study by Kolny [27]. In Warsaw, the capital of Poland, among the programs implemented during the pandemic era, there were initiatives such as “Aktywny warszawiak” (Active Warsaw) and “Aktywnie nad Wisłą” (Active on the Vistula River), which organized cyclic recreational activities for residents on the Warsaw section and the Vistula shore, as well as recreational events promoting water sports [22]. Gdańsk offered a cycle of free activities for all residents regardless of age and physical condition under the slogan “Activate Yourself in Gdańsk”, offering 22 different types of activities, which were used (both in person safely and online) by approximately 50,000 participants [23].

The pandemic also made it necessary to allocate resources to other, more contemporary forms of recreation. As of 2019, so-called e-sports, were very popular, and they continued to grow throughout the pandemic. For example, Katowice focused on one sport, organizing the Intel Extreme Masters tournament, unofficially called the world championships in computer games, gathering not only hundreds of thousands of fans and supporters of e-sports but above all, the world’s top professional gamers and major manufacturers of gaming equipment [24]. The interest in this type of pandemic entertainment has been reflected in many studies, as shown by Marta and colleagues [28], despite the negative implications of such entertainment on children [29]. Apart from the active involvement of local communities in sports and recreation, no studies have been conducted in Poland on the issue of sports volunteering during and after the COVID-19 pandemic, which, as indicated by a British study conducted by Power et al. [30], may be an interesting alternative to social activity.

Both the forms of social activity in the field of sport and recreation adopted in Poland and the financial condition of local governments, are consistent with the situation found in the Czech Republic and Slovakia. Research conducted by Nemec et al. [31] indicated that the COVID-19 crisis significantly impacted local government finance in Czechia and Slovakia on both the expenditure and revenue sides of their budgets. Restrictive measures to counteract the pandemic began in both countries at the beginning of March, and almost immediately, they responded to the first cases of the disease by banning cultural and sporting events [31]. Germany also acted similarly,, adjusting its sports and leisure time activities [32]. Similar actions were taken by Great Britain, assuming that sporting events usually rely heavily on interpersonal contact and banning large public events [33].

In addition, it should be emphasized that during the pandemic, spending time outdoors in green and public spaces was often a source of resilience for maintaining well-being, while also allowing social separation. The Fagerholm et al. study found that people increased their outdoor leisure, particularly in locations that provided a wide range of advantages associated with cultural ecosystem services. People discovered that regardless of how actively they engaged in outdoor recreation during the COVID-19 spring, nature contributed significantly to their subjective well-being. To promote the health advantages of nature in urban areas, urban planners should better adapt to various requirements for outdoor recreation [34]. This is yet another of the challenges facing local governments.

The issue of sports and recreation spending in Poland is very important. The physical activity of the population in Poland is declining. In 2021, one in four of Poland’s residents (28 percent) engaged in sports on a regular basis and with some regularity; in the EU-28, the percentage is closer to 40 percent. More than half of Poland’s residents say they do not play sports at all (56 percent). This means that physical inactivity has increased by 10 percentage points since 2004. The cost of inactivity is estimated at PLN 7 billion a year. Three out of four Europeans believe that opportunities for physical activity are plentiful (74 percent); in Poland, 68 percent hold the same view. Poles are more likely (53 percent) than other EU citizens (39 percent) to believe that local governments are not doing enough to provide opportunities for residents to be physically active [35].

Health departments of local governments in Poland are working with public housing authorities, redevelopment organizations, and city planning departments to promote land-use practices and policies that support physical activity to reduce inequalities in physical activity and related chronic diseases (e.g., protected parks and other green spaces, pedestrian friendly neighborhoods and bike friendly roads) [35].

Research plays a key role in this endeavor, offering compelling health statistics that can be used to influence urban policies to encourage communities to be physically active. These communities can seek funding opportunities for this work and promote a more equitable distribution of resources with the help of findings from scientific publications. While there are many opportunities for municipal health departments in Poland to promote exercise, there are significant barriers to overcome. Since there is a shortage of public money for exercise promotion, efforts should be made to increase funding streams for public health so that this work can be expanded. The potential for increased efficiency and better coordination of physical activity promotion initiatives with other public health programs would be a significant benefit of this flexibility. From the researchers’ perspective, it is important to observe the phenomenon presented in the study, especially the data that relate to the quality of physical activity in the population. Therefore, the study’s authors plan to expand the study after acquiring more data to provide a broader perspective of the statistical variability presented in the research material.

## 5. Strengths and Limitations

Ongoing research relating to spending on sport and recreation in an international and national context may inspire further researchers to expand the subject and conduct, for example, research on the impact of the COVID-19 pandemic on local government spending on recreation and sports services in smaller Polish cities, counties, or even municipalities. How the administrators of individual local government units approach sport in terms of financial support for various activities may bring about completely different and hitherto unknown implications that will cast the subject in entirely new perspectives.

An important limitation of the study is that the analyses focus only on European Union countries and relate them to local issues. In subsequent analyses, the authors aim to conduct a broader scientific investigation examining the impact of financing recreational and sporting activities during a pandemic in other countries of the world. In addition, the use of a previously unused tool measuring residents’ opinion on the performance of local governments in promoting physical activity (sports and recreation) can be counted as a limitation.

The lack of data for 2021 can also be considered a limitation of the study; however, the data for 2020 was published by Eurostat in February 2022, so it is projected that the data for 2021 will be published in the corresponding period in 2023. The authors plan to continue research on the subject as more recent statistics become available.

## 6. Conclusions

From 2015 to 2019, sports and recreation funding were an effective tool for the implementation of measures to develop physical activity in society. They increased the diversity and accessibility of sports and recreational offerings for residents. Funding by local governments for physical recreation and sports services increased steadily until 2019, with a decrease in funding for such activities during the COVID-19 pandemic (2020). A similar trend has also been observed in most European Union countries. Therefore, it can be concluded that Poland did not stand out in this regard and in some aspects even tried to maintain the financial side of social activities at an appropriate level. For example, in Podlaskie and Lower Silesia voivodships, a significant increase in financing was observed in the pandemic period in comparison to the pre-pandemic period.

In addition, it was shown that the opinion of respondents mostly coincided with the existing financial state. In voivodeships where there had been a decrease in spending related to sports and recreation compared to the pre-pandemic period, residents gave local governments lower ratings for the promotion of physical activity.

## Figures and Tables

**Table 1 ijerph-19-12798-t001:** Local government expenditure on recreational and sporting services in European Union countries (in million euros and as the percentage of the total).

Country/Time	Pre-Pandemic										Pandemic		2019–2020 Change (%)	
	2015		2016		2017		2018		2019		2020			
	EUR	%	EUR	%	EUR	%	EUR	%	EUR	%	EUR	%	EUR	%
France	11,664.0	4.7	11,431.0	4.6	11,879.0	4.7	12,096.0	4.7	13,069.0	4.8	12,167.0	4.5	−6.9	−0.3
Spain	3756.0	5.8	3765.0	5.8	4040.0	5.9	4286.0	6.1	4620.0	6.2	4450.0	6.2	−3.7	0
Netherlands	3743.0	3.9	3656.0	3.8	3666.0	3.8	3887.0	3.8	4043.0	3.8	4143.0	3.7	+2.5	−0.1
Sweden	2231.8	2.0	2354.9	2.0	2352.1	2.0	2392.6	2.0	2559.7	2.1	2705.5	2.2	+5.7	0.1
Poland	1524.7	2.7	1333.2	2.4	1661.5	2.7	2102.6	3.0	2103.5	2.8	1886.8	2.4	−10.3	−0.4
European Union *	1266.5	3.8	1247.4	3.8	1316.5	3.9	1408.8	4.1	1494.7	4.2	1438.6	4.0	−3.8	−0.2
Belgium	1311.4	4.5	1404.0	4.7	1423.2	4.6	1658.2	5.1	1625.3	4.9	1430.8	4.3	−12.0	−0.6
Italy	1170.0	0.5	1089.0	0.5	1131.0	0.5	1215.0	0.5	1336.0	0.5	1367.0	0.5	+2.3	0
Finland	916.0	1.9	950.0	2.0	1066.0	2.2	1211.0	2.4	1261.0	2.4	1190.0	2.2	−5.6	−0.2
Denmark	962.9	1.0	983.7	1.0	997.9	1.0	1031.2	1.0	1015.7	1.0	1051.8	1.0	+3.6	0
Greece	482.0	7.9	577.0	9.3	619.0	9.9	696.0	10.5	713.0	11.5	767.0	11.8	+7.6	+0.3
Czechia	484.3	2.5	484.6	2.7	582.2	2.8	732.3	3.0	713.4	2.7	685.1	2.4	−4.0	−0.3
Portugal	520.6	4.9	436.5	4.1	493.4	4.3	578.1	4.9	610.2	5.0	634.2	4.7	+3.9	−0.3
Romania	332.5	2.1	281.8	1.8	329.3	2.0	397.7	2.4	468.7	2.5	458.4	2.2	−2.2	−0.3
Hungary	233.6	2.7	225.3	3.3	316.5	4.0	345.8	4.0	477.5	4.8	376.7	4.3	−21.1	−0.5
Luxembourg	218.5	9.3	237.4	9.2	265.8	9.9	285.0	9.9	296.0	9.8	276.4	8.7	−6.6	−1.1
Croatia	209.7	3.8	182.4	3.3	145.9	2.5	160.9	2.5	179.6	2.6	180.6	2.5	+0.6	−0.1
Lithuania	34.6	1.2	44.4	1.5	66.8	2.0	80.0	2.2	103.1	2.5	124.8	2.6	+21.0	+0.1
Slovakia	102.8	1.7	97.6	1.8	105.0	1.8	127.6	2.0	136.9	1.9	124.0	1.7	−9.4	−0.2
Slovenia	104.5	3.0	93.8	2.8	111.7	3.2	135.1	3.5	126.7	3.1	120.0	2.9	−5.3	−0.2
Ireland	89.9	1.7	92.2	1.6	75.6	1.2	120.2	1.6	122.3	1.4	116.1	1.3	−5.1	−0.1
Estonia	58.5	3.1	72.4	3.7	98.6	4.3	101.2	4.1	120.0	4.3	115.6	4.0	−3.7	−0.3
Bulgaria	172.1	3.6	71.1	2.1	79.8	2.2	74.1	1.8	71.1	1.6	69.1	1.5	−2.8	−0.1
Latvia	41.0	1.7	46.6	1.8	60.7	2.0	64.2	1.9	63.8	1.9	48.8	1.5	−23.5	−0.4
Cyprus	30.4	10.6	27.4	10.5	29.9	10.6	33.4	10.5	36.8	10.6	37.0	10.6	+0.5	0
Malta	--	--	--	--	--	--	--	--	--	--	--	--	--	--
Germany	--	--	--	--	--	--	--	--	--	--	--	--	--	--
Austria	--	--	--	--	--	--	--	--	--	--	--	--	--	--

-- not available; * average for countries with available data.

**Table 2 ijerph-19-12798-t002:** Local government expenditure on physical culture in Poland by region in 2017–2020 (total in million euros and per capita in 2020).

Region/Time	Number of Residents	Pre-Pandemic			Pandemic	2019–2020 Change (%)	per Capita
		2017	2018	2019	2020		2020
Poland	38,265,013	27.29	37.88	40.40	42.53	+5.27	1.11
Lower Silesia	2,891,321	5.16	6.48	4.83	8.50	+75.91	2.94
Kujawsko-Pomorskie	2,061,942	1.15	1.11	1.55	1.61	+4.23	0.78
Lubelskie	2,095,258	1.00	1.03	1.07	1.15	+7.47	0.55
Lubuskie	1,007,145	1.62	1.67	3.67	2.05	−44.27	2.03
Lodzkie	2,437,970	1.76	2.65	2.28	1.94	−14.67	0.80
Malopolskie	3,410,441	2.04	4.31	4.50	4.68	+3.84	1.37
Mazowieckie	5,425,028	1.12	3.94	4.20	4.01	−4.61	0.74
Opolskie	976,774	1.07	1.36	1.33	1.32	−0.22	1.35
Podkarpackie	2,121,229	0.66	0.69	0.73	0.73	−0.54	0.34
Podlaskie	1,173,286	0.77	0.63	0.80	1.49	+86.68	1.27
Pomorskie	2,346,671	1.12	1.22	1.19	1.08	−9.25	0.46
Upper Silesia	4,492,330	5.37	6.87	7.76	8.30	+6.90	1.85
Swietokrzyskie	1,224,626	0.51	0.51	0.59	0.45	−23.91	0.37
Warminsko-Mazurskie	1,416,495	0.66	0.64	0.87	0.85	−1.64	0.60
Wielkopolskie	3,496,450	2.30	3.72	3.85	3.19	−17.00	0.91
Zachodniopomorskie	1,688,047	0.98	1.06	1.18	1.17	−0.10	0.70

**Table 3 ijerph-19-12798-t003:** Relationship between the opinion on the realized activity of local governments in physical activity and the difference in spending for this purpose in the years before and during the COVID-19 pandemic (*n* = 1600).

Region	My Local Government Is Completely up to the Task of Promoting Physical Activity among Residents.	Physical Activity Activities Carried out by My Local Government Are Well Advertised, and Information about Them Is Readily Available.	The COVID-19 Pandemic Has Affected the Number of Physical Activity Services Provided by the Local Government.	COVID-19 Has Affected the Quality of Services Provided by the Local Government in the Field of Physical Activity.	During the COVID-19 Pandemic, Special Undertakings Were Introduced to Promote Physical Activity among Residents.	Average Score	2019–2020 Change (%)	Χ^2^	V	*p*-Value
Lower Silesia	4.56	4.02	4.44	4.12	4.16	4.26	+75.91	10.785	0.651	0.0001
Kujawsko-Pomorskie	4.32	3.98	4.20	4.02	4.04	4.11	+4.23	11.543	0.589	0.0001
Lubelskie	4.12	3.64	4.08	4.24	4.00	4.02	+7.47	9.784	0.678	0.0001
Lubuskie	3.82	2.30	2.36	2.13	3.03	2.73	−44.27	10.643	0.623	0.0001
Lodzkie	3.92	2.46	2.56	2.61	2.98	2.91	−14.67	11.543	0.533	0.0001
Malopolskie	4.04	3.68	3.98	3.08	3.56	3.67	+3.84	12.543	0.486	0.0001
Mazowieckie	4.06	2.34	2.42	2.38	1.98	2.64	−4.61	12.786	0.631	0.0001
Opolskie	3.88	1.98	1.98	1.78	1.78	2.28	−0.22	10.843	0.601	0.0001
Podkarpackie	3.88	2.02	1.98	1.60	1.98	2.37	−0.54	11.864	0.503	0.0001
Podlaskie	3.34	4.86	4.56	4.52	4.68	4.32	+86.68	9.786	0.498	0.0001
Pomorskie	3.24	2.62	2.88	3.04	2.80	2.95	−9.25	31.328	-	-
Upper Silesia	3.82	3.44	3.08	3.04	4.08	3.35	+6.90	9.435	0.456	0.0001
Swietokrzyskie	3.22	1.98	1.78	1.64	2.34	2.16	−23.91	10.764	0.698	0.0001
Warminsko-Mazurskie	3.46	2.68	1.86	1.80	2.06	2.45	−1.64	10.463	0.563	0.0001
Wielkopolskie	4.02	2.22	1.56	1.52	2.14	2.33	−17.00	13.531	0.456	0.0001
Zachodniopomorskie	3.06	2.84	2.96	3.08	3.02	2.99	−0.10	29.874	-	-

## Data Availability

Local data + EUROSTAT.

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
