# Peer review of "Local Governments Spending on Promoting Physical Activity during 2015–2020: Financial Data and the Opinion of Residents in Poland"

_ijerph, 2022, doi:10.3390/ijerph191912798_

Round 1
Reviewer 1 Report (New Reviewer)
Explanation should be provided on what voivodeships are and defined within the introduction.
Given the figures only cover 2020 and 9 months of restrictions is it really realistic to state that the paper is covering spending over COVID.
Greater emphasis needs to be linked in the discussion with the implementation of sports and recreation tasks in Poland
Author Response
Dear Reviewer,
Thank you for taking the time to review the manuscript. Please be advised that any suggested changes have been made and marked in the current version of the manuscript using the color red.
In detail:
- Explanation should be provided on what voivodeships are and defined within the introduction - the meaning of the term 'province' in the introduction has been clarified (line number 92-93)
- Given the figures only cover 2020 and 9 months of restrictions is it really realistic to state that the paper is covering spending over COVID - The available data goes back only to 2020. We intend in the next paper to deepen the analysis (https://appsso.eurostat.ec.europa.eu/nui/submitViewTableAction.do)
- Greater emphasis needs to be linked in the discussion with the implementation of sports and recreation tasks in Poland - the discussion has been supplemented with the required data (line number 319-345)
Greetings!
Reviewer 2 Report (New Reviewer)
I read with interest the paper by Sobczyk et al. with the aim to assess the government involvement in the implementation of sports and recreation tasks in Poland, comparing it to other European Union countries. For this scope, the data regarding expenditures of local government units on collected in the Statistical Office of the European Union for 2015-2020 was used.
The article is well written. The methodologies described are clear, but I suggest extending the analysis of the data to the year 2021 to further state what is described in the article.
Major:
The COVID-19 pandemic originated in Wuhan, China, in December 2019. The spread of the virus in Europe began with isolated cases in France (24 January 2020) and in Italy (31 January 2020). Italy was the first country in the Western World suffering a huge impact by the COVID-19 pandemic and the first to implement large-scale anti-contagion policies after China (March 2020).
The authors use the year 2020 as an example to show the effects of the pandemic on local government expenditure on recreational and sporting services. In my opinion, 2020 cannot be used as a full pandemic year because many of the countries examined did not have a real impact on their policies until after spring 2020 (certainly after March 2020). Moreover, the effects of changes to this expenditure are seen several months later than planned, as they are the result of previous plannings. It would be fair to analyse the year 2021 as a true example of the impact of the pandemic on recreational and sporting services.
Minors:
The article mainly addresses the Polish situation and compares it to the European situation. In my opinion is necessary to emphasise this fact in the title.
I suggest reducing the number of tables, compacting some of the data, to make them easier to understand.
Please revise English (i.e. line 68 - economic economy).
Author Response
Dear Reviewer,
Thank you for taking the time to review the manuscript. Please be advised that any suggested changes have been made and marked in the current version of the manuscript using the color red.
- I suggest extending the analysis of the data to the year 2021 to further state what is described in the article - The available data goes back only to 2020. We intend in the next paper to deepen the analysis (https://appsso.eurostat.ec.europa.eu/nui/submitViewTableAction.do).
- The article mainly addresses the Polish situation and compares it to the European situation. In my opinion is necessary to emphasise this fact in the title. - title modified (line number 1-4).
- Please revise English (i.e. line 68 - economic economy) - (line number 68)
Greetings!
Reviewer 3 Report (New Reviewer)
In table 2, the change for countries with constant values should be zero, not a dash line.
I understand that the local government involvement and people’s opinions matter. But I think you could motivate the paper as why is it important for, let’s say, policymakers? Why should we study this? Are higher involvements a good sign? And why?
Related to my previous comment, why do we expect to observe different results in Poland versus rest of Europe? Is it because of structural institutional differences? You could add a paragraph explaining this.
Author Response
Dear Reviewer,
Thank you for taking the time to review the manuscript. Please be advised that any suggested changes have been made and marked in the current version of the manuscript using the color red.
- In table 2, the change for countries with constant values should be zero, not a dash line. - (line number 170).
- I understand that the local government involvement and people’s opinions matter. But I think you could motivate the paper as why is it important for, let’s say, policymakers? Why should we study this? Are higher involvements a good sign? And why? Related to my previous comment, why do we expect to observe different results in Poland versus rest of Europe? Is it because of structural institutional differences? You could add a paragraph explaining this. - the discussion has been supplemented with the required data (line number 319-345).
Greetings!
Round 2
Reviewer 2 Report (New Reviewer)
Only some of the suggestions have been considered. I still suggest reducing the number of tables and further revising the text editing.
Author Response
Dear Reviewer,
Thank you very much for the suggestions indicated. Regarding them, we would like to inform you that we have decided to reduce (condense) the number of tables in the work. Currently, there are 4 tables in the manuscript, which indicate the most important results - the former tables 1 and 2, and 3 and 4, which actually could have been unreadable and indicate similar data, have been combined. The descriptions of the results were also corrected so that they logically relate We hope that in its current form the work is correctly structured.The descriptions of the results have also been corrected so that they logically relate to the new tables. We hope that in its current form the work is correctly structured.
Thank you and best regards!
This manuscript is a resubmission of an earlier submission. The following is a list of the peer review reports and author responses from that submission.
Round 1
Reviewer 1 Report
The revisions do not meet my expectations. The research design is the same except that some simple tests have been added. The work is largely descriptive and focuses on the interpretation of aggregate statistics. It is difficult to justify the restriction to governmental recreation and sports services, as clubs and non-profit organisations and sponsors are active in this field. The sample is too small to conduct t-tests. The minimum number of observations for a given cross-section is 30. The time period is too short to study the impact of the pandemic on government spending. Decisions on the government budget are made in the previous year. The decimal separator in English is the full stop, not the comma.